# How Structural Variations Influence Crop Improvement

**DOI:** 10.3390/ijms26146635

**Published:** 2025-07-10

**Authors:** Xiaomei Wang, Changyuan Liu, Xiaohuan Sun, Guohong Sun, Chunmei Zong, Yuxin Qi, Yanfeng Bai, Wen Li, Fanjiang Kong, Haiyang Li, Yanping Wang

**Affiliations:** 1Mudanjiang Branch of Helongjiang Academy of Agricultural Sciences, Mudanjiang 157000, China; wxmaiwyp2006@126.com (X.W.); mdjnkylcy@163.com (C.L.); xaozm@yeah.net (X.S.); sunguohong@163.com (G.S.); zongcm@126.com (C.Z.); 6402296@163.com (Y.Q.); mdjnkybyf@163.com (Y.B.); liwenlevi@163.com (W.L.); 2Guangdong Key Laboratory of Plant Adaptation and Molecular Design, Innovative Center of Molecular Genetics and Evolution, School of Life Sciences, Guangzhou University, Guangzhou 510006, China; kongfj@gzhu.edu.cn; 3Yazhouwan National Laboratory, Sanya 572000, China

**Keywords:** structural variation, gene expression, crop improvement

## Abstract

Research on structural variations in the field of crop genetics has expanded with the rapid development of genome sequencing technologies. As an important aspect of genomic variations, structural variations have a profound impact on the genetic characteristics of crops and significantly affect their key agronomic traits, such as yield, quality, and disease and stress resistance—by changing the gene arrangement order, copy number, and the positions of regulatory elements. Compared with single-nucleotide polymorphisms, structural variations present a diverse range of types, including deletions, duplications, inversions, and translocations, and their impacts are more extensive and profound. However, research on structural variations in crops still faces many challenges, for example those relating to different ploidy levels, genome repetitiveness, and their associations with phenotypes. Nevertheless, breakthroughs in long-read sequencing technologies and the integration of multi-omics data offer hope for solving these problems. A deep understanding of the impact of structural variations on crops is of great significance for accurately analyzing the evolutionary history of crops and guiding modern crop breeding, and is expected to provide strong support for global food security and the sustainable development of agriculture.

## 1. Introduction

As the source of biological diversity, genetic variations are an indispensable element for the exploration and development of modern genetics. Our understanding of genetic variation is continuously deepening with the advancement of sequencing technologies and data analysis methods, thereby providing momentum for innovation and discovery across various fields. Genetic variations encompass several key types that contribute to the diversity within a species, including single-nucleotide polymorphisms (SNPs), insertions/deletions (InDels), simple sequence repeat polymorphisms, and structural variations. Among these, SNPs and small InDels have frequently been investigated and used to dissect the genetic basis of key agronomic traits and guide breeding practices [1,2]. Recently, advances in sequencing and computational algorithms have enabled the genome-wide identification of complex structural variations, generally defined as large-fragment deletions or insertions, copy number variations, inversions, and translocations between chromosomes of different individuals [3]. These variations can significantly impact key traits as they can lead to gene loss, duplication, and generation.

Early research on structural variations mainly focused on the human genome, and a series of structural variations associated with human diseases were detected [4,5]. However, the identification of structural variations in plant genomes is challenging due to their size and complexity [6]. Although the efficiency, accuracy, and cost of identification regarding structural variations are all being improved, there are still many urgent problems to be solved in the field of crop research. For example, the genomic structures of different crops vary greatly, making it difficult to widely apply universal methods for identifying structural variations. Even though structural variations have been widely identified in crops, their mechanisms of impact on the specific physiological processes and agronomic traits of crops are difficult to clarify. The study of the frequency and distribution patterns of structural variations at the crop population level has also progressed slowly due to factors such as large sample sizes and complex genetic backgrounds [7]. Despite these challenges, the prospects for progress in crop structural variations are promising. Recent breakthroughs in long-read sequencing technologies, such as PacBio sequencing [8] and Nanopore sequencing [9], offer higher-resolution insights into crop genomes. These technologies can span large repetitive regions that are difficult to sequence with short-read methods, greatly facilitating the detection of complex structural variations. The integration of multi-omics data, including transcriptomics, proteomics, and metabolomics, has also emerged as a powerful approach to better understand the functional consequences of structural variations in crops.

A deep and comprehensive understanding of the impact of structural variations on crops, the accurate identification of structural variations closely related to beneficial traits, and, on this basis, the full use of advanced technologies such as molecular marker-assisted selection and gene editing can greatly accelerate the breeding process of varieties. This can significantly increase crop yields to meet the growing global food demand. This review summarizes the research progress on the impact of structural variations on crops, deeply analyzes the key problems faced in current research, and looks ahead to the promising research directions in the future.

## 2. The Types and Detection of Structural Variations

Structural variations are defined as alterations in the length, copy number, orientation, or chromosomal location of DNA segments among different individuals within the same species. Structural variations mainly encompass six basic types (deletions, insertions, duplications, copy number variations, inversions, and translocations) and involve DNA segments typically larger than 50 base pairs (Figure 1) [10]. Such changes have the potential to impact gene functions, expression levels, or regulatory elements, which may consequently result in phenotypic differences or diseases. Insertions involve the addition of nucleotides into a DNA sequence, while deletions involve the removal of nucleotides. Small insertions or deletions, which are typically less than 50 base pairs in length, occur within coding regions or promoters. They can cause frameshift mutations or the abnormal expression of genes [11]. In contrast, larger insertions or deletions—which may encompass entire genes or chromosomal segments—may have more profound impacts on genome structure and function [12]. Duplication involves the replication of specific DNA segments in the genome, leading to an increase in their copy numbers. They can be categorized into two types: tandem duplications and dispersed duplications. To some extent, copy number variation can be regarded as a type of duplication. It can cover a relatively large genomic region, ranging from changes involving a few genes to those encompassing multiple gene clusters or even chromosomal segments. It therefore has the potential to significantly impact the functions of multiple genes, which has been confirmed in plants [13] and humans [14]. Inversion refers to a structural change in which a segment of DNA on a chromosome is reversed in orientation. Inversions can be classified into two types depending on whether the inverted region includes the centromere: pericentric inversions and paracentric inversions. Translocation refers to the movement of a chromosomal segment from one position to another. This movement can occur within the same chromosome (intra-chromosomal translocation) or between different chromosomes (inter-chromosomal translocation). Translocation may have significant impacts on gene function, chromosomal stability, and the phenotype of an organism.

Due to the limitations of early technologies, structural variations were defined mainly as large-fragment genomic alterations of >3 Mb that could be detected at the microscopic level [4]. About 20 years ago, the advent of hybridization-based microarray approaches and SNP arrays significantly enhanced the feasibility and efficiency of detecting structural variations [15,16]. These methods are now mostly used in fields such as the diagnosis of genetic diseases, genetic analysis of tumors, and population genetics research, and they have aided the discovery of genomic structural variations associated with human diseases. Minute deletions and duplications, which may lead to various congenital diseases, were detected in the fetal genome using hybridization-based microarray approaches [17]. With the advances in detection methods in recent years, especially DNA sequencing technologies and algorithms, whole-genome analysis has become viable for a wide range of plant species, and an increasing number of structural variations have been identified with higher efficiency. Initially, single-end and paired-end reads were employed for DNA sequencing [18]. However, owing to the limitations of these technologies, the sequencing reads were so short that structural variations could not be accurately detected. The latest breakthroughs in long-read sequencing and high-throughput chromosome conformation capture technologies present viable solutions to certain challenges related to short sequence reads [19]. The lengths of sequences obtained using long-read sequencing technologies, such as synthetic long-read sequencing and single-molecule long-read sequencing, can reach 10 to >100 kb [20,21,22]. These technologies have greatly improved the detection efficiency of structural variations. Particularly in recent years, with the development of pan-genomics and telomere-to-telomere (T2T) genomics, the genomes of many crops have been sequenced, like rice, maize, soybean, sorghum, and wheat. Based on these data, numerous structural variations have been detected and confirmed to be associated with traits [23,24]. Wei et al. sequenced two sorghum T2T genomes using ultra-long reads and 52 large inversions (>50 kb), and 68 translocations were detected between two sorghum T2T genomes [25]. The high-quality genomes of 17 representative wheat varieties from more than 5000 cultivars bred in China were assembled, and nearly 250,000 structural variations were identified. Among these, structural variations in the *VRN-A1* and *Ha* genes were found to promote the adaptability and grain hardness of wheat, respectively [26].

## 3. The Naturally Occurring Causes of Structural Variations in Plants

Structural variations can be generated through multiple mechanisms, including non-allelic homologous recombination (NAHR), non-homologous end joining (NHEJ), fork stalling and template switching (FoSTeS), and transposable elements (TEs). NAHR is a biological mechanism for repairing broken chromosomes, which results in gross genomic rearrangements. This mechanism involves recombination between non-allelic homologous sequences. These homologous sequences, which are often repetitive elements such as low-copy repeats or transposable elements, can misalign during meiosis. When recombination occurs between these misaligned sequences, it can lead to the deletion or duplication of the DNA segment between the recombination sites [27]. The size of the deletion or duplication is largely dependent on the positioning of misaligned repeats and even translocations and chromosome fusions [28].

NHEJ is one of the important DNA repair methods for joining double-strand breaks in DNA. Under normal circumstances, it can connect broken DNA ends to maintain the integrity of the genome [29]. DNA double-strand breaks are induced when cells are exposed to physical factors like radiation, chemical factors such as specific mutagens, or biological factors, including viral infections. The NHEJ mechanism is likely to be activated and connect the broken ends. However, if the mechanism is abnormal, it may connect DNA ends incorrectly; for instance, during the repair of multiple DNA double-strand breaks, NHEJ may wrongly connect the broken ends of different chromosomes, thus leading to structural variations such as chromosomal translocations [30].

FoSTeS was proposed by Zhang F in 2009 to explain the genomic rearrangements associated with human diseases [31]. During DNA replication, the replication fork can stall due to various obstacles, such as DNA damage, tightly bound proteins, or sequences that are difficult to replicate. When the replication fork stalls, the replication machinery may switch to a different, nearby DNA template to continue replication. Depending on the location and nature of the template switch, this behavior can lead to the duplication or deletion of DNA segments.

TEs are DNA sequences in the genome that can move autonomously, and they are classified into two major categories based on their movement mechanisms: DNA transposons and retrotransposons. DNA transposons move via a “Cut-and-Paste” mechanism, relying on transposase to recognize and cleave the inverted repeat sequences at both ends of the transposon, excising it from the original locus and inserting it into a new genomic site. In contrast, retrotransposons mobilize via an RNA intermediate. Insertion of the element near or within a gene may disrupt the gene or lead to gene duplication. The transposable elements are widespread in plants and were also first discovered in them [32,33]. They result in the lineage-specific genome expansion and chromosome rearrangements after the split of two plant relatives [34].

## 4. The Impacts of Structural Variations

### 4.1. Crop Growth and Development

In rice, the “Green Revolution” gene *SD1* encodes a key enzyme in the gibberellin synthesis pathway (Table 1). When a 383 bp deletion occurs in *SD1*, the synthesis of gibberellin in rice plants decreases, resulting in plant dwarfism [35]. Dwarf rice varieties have stronger lodging resistance, which is beneficial for increasing yields and has been widely used in rice breeding. An insertion of Ty1/Copia-like retrotransposon leads to the mutation of *PH13* in soybean, which affects the gibberellin content, ultimately resulting in a reduction in plant height [36]. A major gene controlling plant height in upland cotton, *GhPH1*, was cloned and encodes gibberellin 2-oxidase 1A, a 1133 bp structural variation located approximately 16 kb upstream of *GhPH1* that influences the expression of *GhPH1*, thereby affecting plant height [37]. The wheat dwarfing gene *Rht-D1c* leads to a reduction in plant height due to an increase in the copy number, and its dwarfing ability is more than three times that of the single copy gene [38].

Some research has reported that structural variations can impact the development of roots or nodulation. Zhang reported that the insertion of *GmSINE1* transposons results in the loss of function of a specific *R* gene [45]. This change promotes the recognition of native soil rhizobia by soybeans and enhances the nitrogenase activity of nodules as well as the aboveground biomass. The type-II chalcone isomerase genes, unique to leguminous plants, have experienced substantial structural differentiation, which is likely the result of tandem duplication events. This genetic diversification has played a significant role in enhancing the nodulation process in soybeans and *Medicago truncatul* [46]. Transposable elements were found in maize and have influenced many traits. Many allelic variations related to flowering time are caused by transposon insertions [65]. The transposons of *ZmCCT9* and *ZmCCT10* arose sequentially following domestication and were targeted by selection as maize spread from the tropics to higher latitudes [53]. The insertion of a Ty1/copia-like retrotransposon into the *E4* gene led to a decrease in the expression level of the *E4* gene. As a result, soybeans became insensitive to long day conditions [47]. The deletion of the *E1* gene, a core gene associated with flowering time, could significantly advance the flowering time of soybeans [48,49].

### 4.2. Crop Yield

Structural variations are not only associated with agronomic traits but also contribute to crop yield. For example, the gain number is an important trait for rice yield. Structural variations in *Gn1* disrupt its function, resulting in cytokinin accumulation in inflorescence meristems and an increase in the number of reproductive organs [39]. A 1.2 kb transposon-containing insertion 60 Kb downstream of *KRN4* regulated the level of its expression, resulting in kernel row number in maize [54]. A large number of structural variations in tomato were identified through super-pangenome analyses. Among them, a 244 bp deletion detected in exon of *cytochrome P450* may affect plant architecture and yield [61].

Moreover, structural variations have the potential to exert an influence on the diverse alternative splicing patterns of *AUXIN RESPONSE FACTOR 2* transcripts, subsequently contributing to the processes of pod development and size selection in peanut [60]. The size and shape of tomato fruits are important appearance quality traits. Through research on the tomato genome, it has been found that multiple structural variations are related to fruit size and shape. For example, a 102 kb inversion at the SUN locus changes the gene expression regulation pattern in this region, causing the tomato fruit to change from round to oval, which significantly affects the appearance quality [62]. An increase in the copy number of genes related to cell division and expansion has been discovered in large-fruited tomato varieties. These structural variations promote the division and expansion of fruit cells, thereby increasing the size of the fruits [66]. In rice, a two-haplotype deletion in the promoter of *GSE5* reduced the expression of *GSE5,* which caused wide and heavy grain. Conversely, the overexpression of GES5 resulted in narrow grains [43].

### 4.3. Crop Quality

Crop quality is important in many fields such as human life, agricultural production, and the ecological environment. Taking the nutritional quality of crops as an example, many studies have revealed the crucial roles played by structural variations. Glucosinolates—an important class of secondary metabolites in rapeseed—not only affect the disease and pest resistance of rapeseed but also are closely related to the quality of rapeseed oil. By integrating population transcriptome data, structural variations can regulate the expression of genes related to the synthesis and transport of glucosinolates in rapeseed [64]. A 1454 bp insertion downstream of *BnaA03.MAMf* regulated the level of its expression, resulting in lower 4C:(4C + 5C) or higher 5C:(4C + 5C) ratios of glucosinolate. Meanwhile, accessions carrying a 41.6 kb insertion in the *BnaA09.MYB28* exhibited significantly higher glucosinolate content compared to those without the insertion [64]. The low-molecular-weight glutenin subunit is an essential part of the storage protein in wheat, and the composition, copy number, and expression of core genes all influence wheat quality traits [67,68].

### 4.4. Crop Stress Resistance

Although plants are highly adaptable to these environments, transient changes in their growth environment still pose stress to them. Structural variations enable plants to adapt to environmental changes, and environmental changes can also give rise to the occurrence of structural variations. Some studies have found that environmental stress can activate transposons. For example, the transcriptional activity of TE-related genes in rice under stress and normal conditions was detected, and it was found that transcriptional activity of TE-related genes was significantly upregulated under stress conditions [69]. Tto1 and Tnt1 retrotransposons in tobacco and oat were activated by physical stresses like cutting and stab wounding, elicitors of plant defense responses, and pathogen infection [70,71].

In barley, an increase in the copy number of the boron transporter gene *Bot1* enables the barley variety Sahara to tolerate boron toxicity [58]. *CBF* (C-repeat binding factor) is located at the frost resistance locus 2 (FR-2) in wheat and barley, and its copy number variations are associated with low-temperature tolerance. The copy number of *CBF14* in winter wheat is higher than that in spring wheat. Deletions of large genes (including *CBF-B12*, *CBF-B14*, and *CBF-B15*) at the FR-2 locus in tetraploid durum wheat and hexaploid bread wheat lead to reduced cold tolerance [56,72,73,74]. In rice, cold can induce the expression of the *LIP19* gene, the high expression level of which is significantly associated with cold tolerance. Interestingly, the 150 bp insertion in the promoter of *LIP19* resulted in significantly higher *LIP19* than normal, which promoted cold tolerance in *Oryza sativa japonica* (with a 150 bp insertion) compared to *Oryza sativa indica* without a 150 bp insertion [41]. He et al. conducted a large-scale screening of the genomic inversion in cultivated and wild rice accessions and found that genomic inversion widely exists [42]. An inversion cluster including *MADS56* showed significantly higher expression levels and stronger heat-resistance compared to the non-inversion cluster. In wheat, an increase in the copy number of *Vrn-A1* prolongs the vernalization time and results in delayed flowering, while a high copy number of *HvFT1* in barley accelerates the flowering time [57,59]. A compelling association was unearthed between the expression of genes implicated in seed oil content and genomic structural variations within the quantitative trait loci of *Brassica napus.* Specifically, an insertion of 6313 bp in the promoter region of the *BnaA09g48250D* gene had a significant connection with the seed oil content [63]. Transposon-mediated inverted repeats (21.4 kb) in maize were found though bioinformatics, genetics, and molecular biology, resulting in environmental adaptation and yield balance [55].

### 4.5. Domestication and Evolution

Structural variations have an interactive and close relationship with crop domestication and evolution, providing an important genetic basis and driving force. In turn, crop domestication and evolution influence the direction and frequency of structural variations and, together, these mutual influences drive the development of crops from a wild state to one that is better adapted to human needs and environmental changes. Structural variations have the capacity to engender novel genetic combinations at the genomic scale. This process serves as a wellspring of rich raw materials, fueling the domestication and evolution of crops. During soybean domestication, the seed coat color shifted from wild soybean black to the most cultivated soybean yellow, and this process was influenced by structural variations. The classical I locus drives this change. Certain structural variation events formed different I locus haplotypes, which are genomic level structural changes. These alter gene expression, diversifying seed coat colors. Different haplotypes, linked to different domestication stages, show that structural variation-induced genetic differences underpin the evolution of soybean seed coat color traits. Humans, based on appearance needs, chose yellow coated soybeans, spurring their evolution into cultivars [47,50,51].

In wheat, copy number variations in the vernalization gene *VRN-A1* play a crucial role in the evolution of spring-type and winter-type wheat. It was found, through the analysis of high-quality genomes of 17 representative wheat varieties, that there is a large block copy number variation of approximately 0.5 Mb in the interval where the *VRN-A1* gene is located. This variation includes structural changes such as duplications and deletions, which directly affect the gene expression level. In cold regions, winter-type varieties with high copy numbers are retained due to their frost resistance advantages, while in warm regions, spring-type varieties with low copy numbers are preferred [26]. Maize was directly domesticated from teosinte (*Zea mays* ssp. *parviglumis*). Before it was confirmed that teosinte was the domesticated ancestor of maize, Tripsacum was thought to be the origin of maize domestication. Through genomic comparative analysis, this study identified the largest (3.2 Mb) variant fragment (named Region A) between the genomes of two inbred maize lines, B73 and Mo17. In the natural population, this fragment showed a high degree of sequence integrity, existing only in two forms: complete presence and complete absence. Through in-depth research on the sequence of this fragment, it was found that there are long terminal repeat retrotransposons specific to *Tripsacum* within it, indicating that the *Tripsacum* genome has contributed to the formation of this region in the maize genome. Signals of gene flow between maize and *Tripsacum* were also detected within this fragment, suggesting that *Tripsacum* may have directly participated in the speciation process of maize [75].

Some studies have revealed that structural variations have an important effect on the domestication and evolution of rice. Qin selected 35 rice materials and constructed a pan-genome using third-generation sequencing technology. Based on these sequences, 171,072 structural variations, 125,889 InDels, 627 inversions, and 4346 translocations were detected. Of them, a 66.6 kb deletion in japonica rice materials and a 43.3 kb deletion in Indica rice materials both contain the complete sequence of the known negative regulatory gene for blast resistance, *OsWAK112d*. Evidently, these two independent deletions contribute to the environmental adaptation of rice by enhancing blast resistance [40].

## 5. How to Apply Structural Variations to Crop Breeding

### 5.1. Molecular Marker-Assisted Breeding

Based on closely linking the molecular marker and target traits, molecular marker-assisted breeding enables accurate selection in the early stages of crop breeding, which helps to accelerate the development of superior varieties and enhances the quality and adaptability of the final varieties. Compared to SNPs, structural variations can be more easily developed into molecular markers and efficiently used for material detection. The soybean cyst nematode is estimated to be the pest that causes the most yield loss in soybean. The copy number of the *Rhg1* gene mediated resistance to soybean cyst nematode. Lee et al. have developed an efficient method for measuring the copy number variation, and have applied this method to improve cyst nematode resistance in soybean [52]. In rice, the copy number variation in *GL7* regulating longitudinal cell elongation was associated with grain size diversity and selected for and used in breeding [44]. Moreover, structural variations can be used to understand the genetic basis of complex agronomic traits. By analyzing the distribution of structural variations in different crop populations, we can gain insights into the evolutionary processes and genetic diversity of crops. This knowledge can guide us to more precisely combine different genetic resources, creating new crop varieties with beneficial, comprehensive traits and better adaptation to various environmental challenges in the future. Shading can cause exaggerated stem elongation in soybean, leading to lodging and reduced yields when planted at a high density. A Ty1/Copia-like retrotransposon insertion in the *PH13* gene results in a truncated PH13 protein, which causes reduced plant height and enables high-density planting [36]. Molecular markers based on retrotransposon insertion in the *PH13* gene were developed, followed by introgression of the insertion-type *ph13* allele into wild-type materials, enabling the rapid genotypic selection of progeny (Figure 2A).

### 5.2. Gene Editing Technology

With the development of gene editing technologies, it is now possible to perform precise modifications at specific locations within the genome, including introducing, repairing, or deleting structural variations. By precisely editing the structural variation regions in crops, researchers can determine the specific impacts of these structural variations on target traits and thereby accurately screen out those with breeding value.

Based on the crop breeding target, researchers can design and create novel structural variations. They can simultaneously edit multiple genes associated with important traits and introduce them into a single crop variety to aggregate beneficial traits. In 2024, researchers introduced a 10 bp heat shock element into tomato and rice through a new editing system, which increased the yield in a heat environment [76]. *GSE5* regulated grain size in rice, and deletion in the promoter of *GSE5* reduced expression and then caused wide grain and increased yield [43]. Gene editing was used to knock out the promoter of *GSE5*, thereby reducing its expression level and rapidly improving grain size (Figure 2B). With the combination of gene editing and structural variation, researchers can rapidly improve the target traits and increase grain yield.

## 6. Opportunities and Challenges

In the field of crop genetics and breeding, structural variations are gradually becoming the focus of research. Structural variations are an important source of genetic diversity in crops. Traditional genetic variations mainly focus on single-nucleotide polymorphisms, while structural variations can generate more complex and extensive genetic changes. These variations can create new gene combinations and alleles, providing rich genetic materials for crop breeding. With the help of modern molecular biology techniques, such as high-throughput sequencing and gene-editing technology, we can quickly and accurately detect and identify structural variations in crops. This enables us to precisely select target traits at the early stage of breeding, greatly shortening the breeding cycle. However, research on structural variations also faces some challenges, for example in terms of their detection and identification. Structural variations typically involve large genomic fragments, and their detection and identification require high-precision technical means. Although great progress has been made in high-throughput sequencing technology, there are still problems of inaccurate detection or missed detection for some complex structural variations, such as hidden inversions and translocations. Moreover, different sequencing platforms and analysis methods may lead to differences in results, which poses a significant challenge to the accurate identification of structural variations. Determining the biological functions of structural variations is an extremely challenging task. Since structural variations may affect multiple genes and their regulatory networks, their impacts on crop traits often exhibit complex pleiotropy. A single structural variation can influence multiple traits simultaneously, and different genetic backgrounds and environmental conditions can also affect its functions. Therefore, to accurately decipher the functions of structural variations, it is necessary to comprehensively apply multidisciplinary methods such as genetics, molecular biology, and bioinformatics and conduct a large number of experimental verifications.

## Figures and Tables

**Figure 1 ijms-26-06635-f001:**
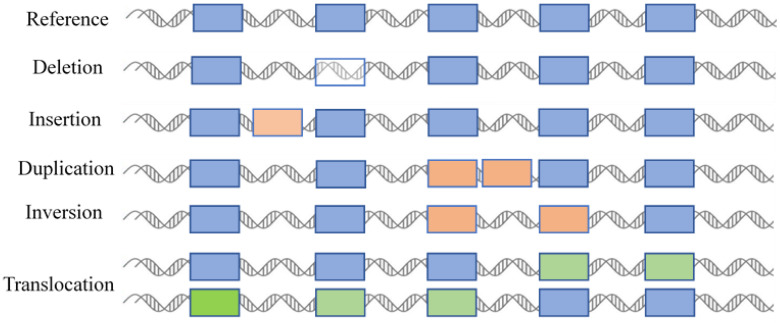
Types of structural variations. Different colored boxes represent different fragments or genes in the genome. >50 bp is used as an operational demarcation.

**Figure 2 ijms-26-06635-f002:**
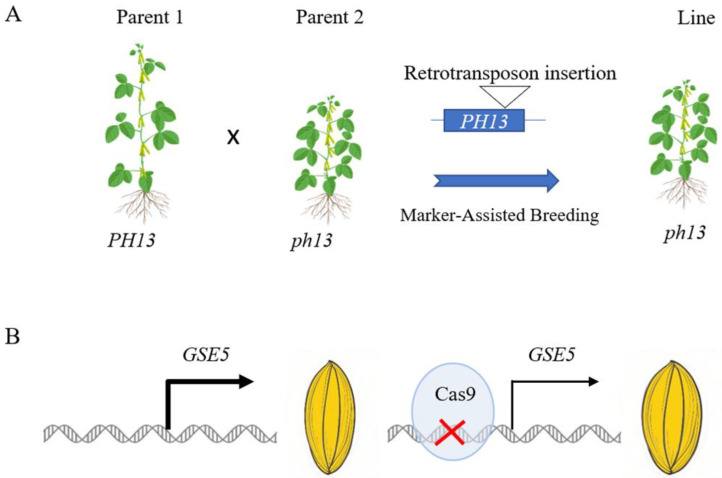
Application of structural variations for crop breeding. (**A**) in a molecular-marker-assisted breeding approach, Parent 1 carrying the wild-type *PH13* allele was crossed with the parent harboring the retrotransposon insertion *ph13* allele. Subsequently, a specific molecular marker was developed and employed to screen the genotypes of the breeding progeny with *ph13* allele. (**B**) using a gene editing-based breeding approach, cultivars carrying the wild-type *GSE5* were employed to edit the promoter of *GSE5*, which reduced the expression of *GSE5* and resulted in wider and heavier seeds.

**Table 1 ijms-26-06635-t001:** Recent structural variations studies in crops.

Species	Gene	Type	Traits	References
Rice	*SD1*	Deletion	Semi-dwarf	Sasaki et al., 2002 [35]
	*Gn1*	Deletion	Grain number	Ashikari et al., 2005 [39]
	*OsWAK112d*	Deletion	Blast disease	Qin et al., 2021 [40]
	*LIP19*	Insertion	Cold tolerance	Li et al., 2024 [41]
	*MADS56*	Inversion	Heat tolerance	He et al., 2024 [42]
	*GSE5*	Deletion	Grain size	Duan et al., 2017 [43]
	*GL7*	Copy number	Grain size	Wang et al., 2015 [44]
Soybean	*PH13*	Translocation	Plant hight	Qin et al., 2023 [36]
	*SINE1*	Insertion	Nitrogenase activity of nodule	Zhang et al., 2021 [45]
	*CHI*	Copy number	Nodule	Liu et al., 2024 [46]
	*E4*	Translocation	Flowering time	Liu et al., 2020 [47]
	*E1*	Deletion	Flowering time	Xia et al., 2012 [48]
	*E1*	Deletion	Flowering time	Zhai et al., 2015 [49]
	*E6*	Translocation	Flowering time	Fang et al., 2015 [33]
	*CHS*	Copy number	Seed coat color	Tuteja et al., 2004 [50]
	*CHS*	Copy number	Seed coat color	Zhou et al., 2015 [51]
	*CHS*	Copy number	Seed coat color	Liu et al., 2020 [47]
	*Rhg1*	Copy number	Soybean cyst nematode resistance	Lee et al., 2016 [52]
Maize	*ZmCCT9*	Translocation	Flowering time	Huang et al., 2018 [53]
	*ZmCCT10*	Translocation	Flowering time	Huang et al., 2018 [53]
	*KRN4*	Translocation	Kernel row number	Liu et al., 2015 [54]
	*ZmMYBR38*	Translocation	Drought response	Sun et al., 2023 [55]
Wheat	*Rht-D1c*	Copy number	Plant height	Tian et al., 2024; Li et al., 2012 [37,38]
	*FR-2*	Copy number	Low-temperature tolerance	Francia et al., 2016 [56]
	*Vrn-A1*	Copy number	Flowering time	Jiao et al., 2025; Armour et al., 2000 [26,57]
barley	*Bot1*	Copy number	Boron toxicity	Sutton et al., 2007 [58]
	*FT1*	Copy number	Flowering time	Nitcher et al., 2013 [59]
Peanut	*ARF2*	Deletion	Seed size	Yin et al., 2019 [60]
Tomato	*Cytochrome P450*	Deletion	Plant architecture	Li et al., 2023 [61]
	*SUN*	Translocation	Grain size	Xiao et al., 2008 [62]
Brassica napus	*BnaA09g48250D*	Insertion	oil content	Zhang et al., 2024 [63]
	*BnaA03.MAMf*	Deletion	glucosinolates ratio	Zhang et al., 2024 [64]
	*BnaA09.MYB28*	Insertion	glucosinolate content	Zhang et al., 2024 [64]
Upland cotton	*GhPH1*	Insertion	Plant height	Tian et al., 2024 [37]

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
