# Peer review of "How Structural Variations Influence Crop Improvement"

_ijms, 2025, doi:10.3390/ijms26146635_

Round 1
Reviewer 1 Report
Comments and Suggestions for Authors
In this review, Wang et al. discussed the types of SV, the molecular mechanisms generating SV, the impact of SV and the application of SV in plant breeding. This appear to be a timely review highlighting the growing interests in SV. However, there are some omissions and writing errors that need to be corrected.
Major:
1, Line 116, when discussing the mechanisms causing SV, TE is a major factor that needs to be discussed.
2, How the development of T2T genome and pan-genome impact SV identification should be included to make the review more relevant to technology advance in the filed, such as the https://onlinelibrary.wiley.com/doi/10.1002/imt2.193.
Minor:
1, reviews have been published in this field, which need to be acknowledged in the manuscript. Such as, https://www.cell.com/molecular-plant/fulltext/S1674-2052(18)30383-6
https://link.springer.com/article/10.1186/S13059-020-02224-8
2, In line 39, Simple Sequence Repeats (SSRs) is not a type of genetic variation
4, Line 43, this is an incorrect sentence, please rephrase
5, Line 47, add “associated” before “with”
6, Line 53, this is an incorrect sentence, please rephrase
Author Response
Dear Reviewer
We have completed the language editing of the manuscript in accordance with the comments from both reviewers and the editor.
Comments and Suggestions for Authors
In this review, Wang et al. discussed the types of SV, the molecular mechanisms generating SV, the impact of SV and the application of SV in plant breeding. This appear to be a timely review highlighting the growing interests in SV. However, there are some omissions and writing errors that need to be corrected.
Major:
1, Line 116, when discussing the mechanisms causing SV, TE is a major factor that needs to be discussed.
Response: Thank you so much for your attention. We have discussed the mechanism of TE causing SV in Line 121and Line 142-146.
2, How the development of T2T genome and pan-genome impact SV identification should be included to make the review more relevant to technology advance in the filed, such as the https://onlinelibrary.wiley.com/doi/10.1002/imt2.193.
Response: Thank you so much for your attention. We have discussed the advantages of long-read sequencing technologies for SVs detection in Line 55-60. And, we have added the discussion on how T2T genome and pan-genome approaches improve SV identification in line 108-116.
Minor:
1, reviews have been published in this field, which need to be acknowledged in the manuscript. Such as, https://www.cell.com/molecular-plant/fulltext/S1674-2052(18)30383-6
https://link.springer.com/article/10.1186/S13059-020-02224-8
Response: Thank you so much for your attention. We have added two references in line 111 and discussed the impact of pan-genome for crop improvement.
2, In line 39, Simple Sequence Repeats (SSRs) is not a type of genetic variation
Response: Thank you so much for your suggestion. We have corrected as“Simple Sequence Repeats Polymorphism”
4, Line 43, this is an incorrect sentence, please rephrase
Response: We have corrected as “Recently, advances in sequencing and computational algorithms have enabled genome-wide identification of complex structural variations, generally defined as large fragment deletion or insertion, copy number variation, inversion and translocation between chromosomes of different individuals’’.
Meanwhile, we also had the language edited by the recommended language editors of MDPI.
5, Line 47, add “associated” before “with”
Response: We have corrected as “Early research on structural variations mainly focused on human genome, and a series of structural variations associated with human diseases were detected”.
6, Line 53, this is an incorrect sentence, please rephrase
Response: We have corrected as “Even though structural variations have been widely identified in crops, their impact mechanisms on the specific physiological processes and agronomic traits of crops are difficult to clarify”. Meanwhile, we also had the language edited by the recommended language editors of MDPI.
Reviewer 2 Report
Comments and Suggestions for Authors
In the current review article, the authors discussed the characteristics and causes of structural variations in plants and the agronomic implications thereof. By enumerating relevant research highlights in the field, the paper also provides insights into the future directions of profiling and harnessing structural variations for improved crop performance. I have a few suggestions:
- It may be helpful to change the subtitle at line 111 from “The generation of structural variations” to something like “The naturally occurring causes of structural variations in plants”, as this section mainly discusses mechanistic grounds for the genetic alterations, rather than man-made genome engineering efforts.
- Lines 135-138: it is worth noting that the transposable element described in reference 28 is not a retroelement, hence not a proper example for retrotransposition.
- “chromothripsis, telomere dysfunction and errors in DNA replication and repair” from line 114 was not elaborated as were other highlighted causes for structural variations. The authors should either remove them from the text or expand on the discussion.
- Line 140: change “impaction” to “impacts”.
- The adopted format for the word “bp” is not uniform across the manuscript. For instance, on line 148, it’s “1133-bp”; on line 213, it’s “150bp”; and on line 222, it’s “6313 base-pairs”. Please standardize.
- Similarly, the use of the hyphen symbol (“-“) needs to be standardized.
- Many of the paragraphs are overly lengthy and can benefit from being broken into multiple smaller chunks of texts. For instance, on line 151, the sentences after “Some research has reported that…” may start as a new paragraph; same for “Moreover, structural variations…” on line 172 and many others.
- Line 301: add a space between “research” and “(Table 1)”.
- Line 546: remove the characters over Fig 1.
Author Response
Open Review
In the current review article, the authors discussed the characteristics and causes of structural variations in plants and the agronomic implications thereof. By enumerating relevant research highlights in the field, the paper also provides insights into the future directions of profiling and harnessing structural variations for improved crop performance. I have a few suggestions:
1.It may be helpful to change the subtitle at line 111 from “The generation of structural variations” to something like “The naturally occurring causes of structural variations in plants”, as this section mainly discusses mechanistic grounds for the genetic alterations, rather than man-made genome engineering efforts.
Response: Thank you so much for your suggestion. We have corrected the “The generation of structural variations” as the “The naturally occurring causes of structural variations in plants”.
2.Lines 135-138: it is worth noting that the transposable element described in reference 28 is not a retroelement, hence not a proper example for retrotransposition.
Response: Sorry, we have added the discussion of transposable element and the reference for retrotransposition.
Fang C, Liu J, Zhang T, Su T, Li S, Cheng Q, Kong L, Li X, Bu T, Li H, Dong L, Lu S, Kong F, Liu B. A recent retrotransposon insertion of J caused E6 locus facilitating soybean adaptation into low latitude. J Integr Plant Biol. 2021 Jun;63(6):995-1003. doi: 10.1111/jipb.13034
3.“chromothripsis, telomere dysfunction and errors in DNA replication and repair” from line 114 was not elaborated as were other highlighted causes for structural variations. The authors should either remove them from the text or expand on the discussion.
Response: Thank you so much for your suggestion. We have removed the "suppress" “chromothripsis, telomere dysfunction and errors in DNA replication and repair”.
4.Line 140: change “impaction” to “impacts”.
Response: We have corrected as“impacts”.
- The adopted format for the word “bp” is not uniform across the manuscript. For instance, on line 148, it’s “1133-bp”; on line 213, it’s “150bp”; and on line 222, it’s “6313 base-pairs”. Please standardize.
Response: Thank you so much for your suggestion. We have standardized "bp" throughout the manuscript.
6.Similarly, the use of the hyphen symbol (“-“) needs to be standardized.
Response: We have standardized "-" throughout the manuscript.
7.Many of the paragraphs are overly lengthy and can benefit from being broken into multiple smaller chunks of texts. For instance, on line 151, the sentences after “Some research has reported that…” may start as a new paragraph; same for “Moreover, structural variations…” on line 172 and many others.
Response: Thank you so much for your suggestion. We have broken the paragraphs, in Crop Growth and Development, Crop Yield, Crop stress resistance and Domestication and evolution.
8.Line 301: add a space between “research” and “(Table 1)”.
Response: We have added a space.
- Line 546: remove the characters over Fig 1.
Response: Thank you so much for your suggestion. We have removed the characters over Fig 1 and corrected the figure legend.
Round 2
Reviewer 1 Report
Comments and Suggestions for Authors
The authors have addressed all my comments